# Comparative Analysis of Building Representations in TELEMAC-2D for Flood Inundation in Idealized Urban Districts

**Zejin Li [1], Jiahong Liu [1,2,3,*], Chao Mei [1], Weiwei Shao [1], Hao Wang [1,2,3] and Dianyi Yan [1]**

[1] China Institute of Water Resources and Hydropower Research, State Key Laboratory of Simulation and Regulation of Water Cycle in River Basin, Beijing 100038, China

[2] School of Transportation and Civil Engineering & Architecture, Foshan University, Foshan 528000, China

[3] Engineering and Technology Research Center for Water Resources and Hydrogeology of the Ministry of Water resources, Beijing 100044, China

* Correspondence: liujh@iwhr.com

**Abstract:** This study aims to better understand the impact of different building representations and mesh resolutions on urban flood simulations using the TELEMAC-2D model in idealized urban districts. A series of numerical models based on previous laboratory experiments was established to simulate urban flooding around buildings, wherein different building layouts (aligned and staggered) were modeled for different building representations: building–hole (BH), building–block (BB), and building–resistance (BR) methods. A sensitivity analysis of the Manning coefficient for building grids indicated that the unit-width discharge and water depth in building grids reduce as the Manning coefficient is less than $10^4$ $m^{-1/3} \cdot s$. The simulated depths via the BH, BB, and BR methods were compared with the measured data in terms of three accuracy indicators: root mean square error, Pearson product–moment correlation coefficient, and Nash–Sutcliffe efficiency. Observing apparent discrepancies based on the hydrographs was difficult; however, some slight distinctions were observed based on the aforementioned three indicators. The sensitivity of 1, 2, and 5 cm mesh resolutions was also analyzed: results obtained using 1 cm resolution were better than those obtained using other resolutions. The complex flow regime around buildings was also investigated based on mesh resolution, velocity, and Froude number according to our results. This study provides key data regarding urban flood model benchmarks, focusing on the effect of different building representations and mesh resolutions.

**Keywords:** urban flood; building representations; TELEMAC-2D model; mesh resolution; comparative analysis

## 1. Introduction

As the global climate has been changing, the number and intensity of natural disasters have significantly increased in recent decades [1,2]. Owing to the frequent occurrence of urban floods caused by heavy rain, urban floods have become an important question and research interest has increased in this field [3]. Cities are widely impacted and heavily damaged by floods because of their vulnerability and sensitivity, particularly in rapidly developing urban areas. For example, Beijing suffered significant damage from a heavy rainstorm in 2012, the losses for which exceeded 1.86 billion USD; more than 1.6 million people were impacted by the resulting flood [4].

Existing research on flood risk has focused on flood modeling, which indicates a lack of field investigation. This is because:

- urban areas are more complex than rural areas owing to more artificial infrastructure;
- receiving precise Digital Elevation Model (DEM) data that records variations in urban micro-topography is difficult;
- related data such as depth and velocity are difficult to observe in fact, and for an entire flow area, data provided via aerial imagery is insufficient.

Furthermore, flood models mainly depend on the quality of the hydrological data input; following this, data must be simulated using numerical models [5]. Hence, urban flooding can be studied via laboratory experiments as they can provide accurate measurements of flow characteristics under controlled conditions. Additionally, laboratory experiments are repeatable and could provide a reliable benchmark for the calibration and evaluation of numerical models. According to a previous study, four major flow types exist in laboratory experiments: (i) flow in street intersections, (ii) vertical exchanges of water, (iii) flow within regular grids of emerging rectangular obstacles and (iv) flow within more realistic urban districts, and observed data from several related experiments have been reported thus far [5].

With the rapid development of urbanization, land use has changed markedly. In urban areas, land use is significantly different from that in natural areas. For example, in urban areas, numerous buildings, roads, and other artificial facilities exist; however, there is a lack of lakes and vegetations. These are important characteristics in urban environments, which should not be ignored in urban flood simulations. Simulating urban flooding in a scientific manner with accurate simulation of urban characteristics is crucial for obtaining accurate results for urban flood models. Buildings are one of the most important components in urban flood modeling and their resistance to water flow is a critical factor in urban flooding. In recent years, flooding around urban buildings has become a major research subject. Liu et al. conducted a laboratory experiment in a large flume to study water-level evolution processes around and within buildings [6]. Fixed obstacles and transportable floating bodies were compared by Albano et al. [7] to analyze the mitigation efficiency of different structural interventions. In addition, porosity has been adopted as a statistical descriptor and introduced into classic shallow water equations (SWEs) to reflect the impact of buildings on urban flooding [8–11]. Wu et al. indicated that increasing bed elevation is still a common method to represent the blockage effect of buildings on surface flow [12]. Zhou et al. defined a blockage factor $\lambda$ and compared the influence of impervious and pervious building blocks in a simplified urban district [13]. Most studies focused on mesh design, model parameterization, and boundary conditions to consider the potential effects of buildings; in contrast, some studies considered urban buildings as a porosity parameter by modifying the classical SWEs [14].

Mignot et al. summarized a lot of experimental tests of idealized urban flooding, which made great contributions to urban flood models [5]. Among these cases, the Toce River Valley has been implemented across fields such as urban flooding research and numerical modeling. Additionally, this case has significantly motivated the development of urban flood modeling and risk management. The case was chosen to study in this research not only because this case is a classical test scenario in urban flood field and in accordance with the study's targets but also because it has been widely adopted around the world and it has been referred to in several research works. Kim et al. indicated that the use of mixed mesh and triangular mesh designs in the Toce case had a comparable level of accuracy [15]. An et al. used the Gerris open source code for SWEs to validate model performance when embedded solid boundaries were included [16]. Ferrari et al. compared isotropic and anisotropic schemes using the Toce case and found that the latter showed better agreement with the measurements than the former [8]. Costabile analyzed the Toce case in terms of model suitability for representing the influence of obstacles on flow propagation [17]. Therefore, the importance of the Toce River Valley case is self-evident. More information about this case is presented in Section 2.2. Herein, we aim to further contribute to urban flood modeling benchmarks using the Toce River Valley case.

The TELEMAC-2D model is a powerful two-dimensional (2D) hydrodynamic model that has been widely used to simulate free-surface flow in various hydrodynamic modeling applications such as dam breaks, harbor structure design, and river floods [18–20]. The source code of

TELEMAC-2D is open and free, which is helpful in developing a model suitable for urban situations and could promote the development of risk pre-warnings in addition to management for urban floods. However, little research has applied this model to urban flooding simulation, indicating that the present study would provide novel and valuable insights in this regard.

Globally, urban flooding is increasingly becoming a significant natural disaster. Numerical modeling is one of the most efficient methods with which to study urban floods. In this study, physical experiments and numerical modeling were performed to analyze the impact and characteristics of urban floods around buildings. We used data from the Toce River Valley case, as obtained from Testa et al. [21], to study different building representation methods and mesh resolutions using the TELEMAC-2D model. The major aims of this study were

- to build TELEMAC-2D models and verify their applicability to urban flooding;
- to compare the results of different building representation methods using numerical modeling;
- to analyze the influence of building layouts and mesh resolutions.

This research is expected to contribute to a deeper understanding of numerical modeling and urban flooding.

## 2. Methods and Materials

### 2.1. TELEMAC-2D Model

The TELEMAC-MASCARET model system, developed by the French National Hydraulic and Environment Laboratory (http://www.opentelemac.org/), is a powerful integrated modeling tool for dealing with one-dimensional (1D), 2D, and three-dimensional (3D) flows [22]. TELEMAC-2D is one of the components of the TELEMAC-MASCARET model system. It solves Saint Venant equations or SWEs based on the finite element method (FEM) over non-structured triangular grids. In the 6th version of TELEMAC-2D, finite volume methods were added to solve SWEs, providing well-balanced scheme properties [23]. This 2D model offers users several classical discretization methods for continuity and momentum equations. The solved equations in their non-conservative form are as follows:

$$\frac{\partial h}{\partial t} + \frac{\partial (hu)}{\partial x} + \frac{\partial (hv)}{\partial y} = 0 \tag{1}$$

$$\frac{\partial u}{\partial t} + u \frac{\partial u}{\partial x} + v \frac{\partial u}{\partial y} = -g \frac{\partial Z}{\partial x} + F_x + \frac{1}{h} \nabla \left( h v_e \nabla u \right) \tag{2}$$

$$\frac{\partial v}{\partial t} + u \frac{\partial v}{\partial x} + v \frac{\partial v}{\partial y} = -g \frac{\partial Z}{\partial y} + F_y + \frac{1}{h} \nabla \left( h v_e \nabla v \right) \tag{3}$$

where $h$ is flow depth; $Z$ is water surface elevation; $u$ and $v$ are depth-integrated velocity components in the $x$- and $y$-directions, respectively; $t$ is time; $g$ is gravitational acceleration; $v_e$ is effective diffusion for turbulent viscosity and dispersion; and $F_x$ and $F_y$ are source terms that ignore the Coriolis force and the influence of wind. Equations (1)–(3) are solved for three main unknowns $h$, $u$, and $v$, and the boundary conditions and meshes are set by the user in the BlueKenue software (https://nrc.canada.ca/en) Full details of the equations can be found in Hervouet's book [22].

### 2.2. Toce River Valley Case

Testa et al. presented a study of a classic physical model of a flash flood in the Toce River Valley. The study was performed under the European Union Investigation of Extreme Flood Processes and Uncertainty (EU IMPACT) project to study flooding in simplified urban-like environments [21]. Constructed in northern Italy, the physical model was 50 m long and built of concrete to a scale of 1:100. It included urban buildings that were represented by concrete cubes measuring 15 cm at the side. The physical model had highly precise terrain data with a 5 cm

resolution, and the water depth data were recorded using several electrical conductivity gauges. The observation equipment recorded the water level at 0.2 s intervals during the experiment, providing a high spatial and temporal resolution for model validation. The gauges did not interfere with the flood flow because they were located above the water surface and supported by spars (see Figure 1).

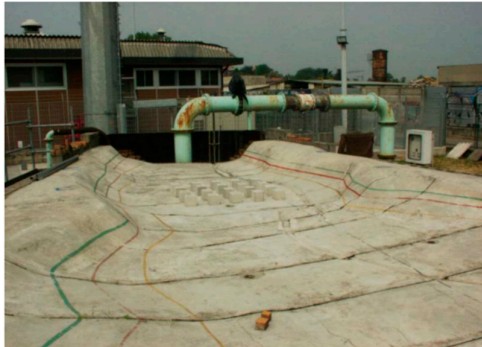 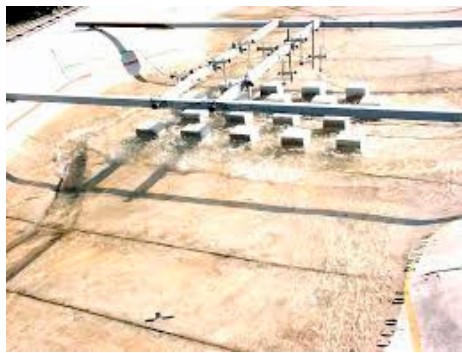

**Figure 1.** Experimental setup of the Toce River Valley case [21].

In this case, only a 5 m-long region close to the upstream area was used. In total, 10 electrical conductivity gauges were used, of which two gauges were located at the entrance of the inflow to obtain the flood discharge, and eight gauges were located in the vicinity of concrete buildings to record the variation in water level over time. The experiment assessed two different topographic configurations (original and modified) and three inflow discharges (low, medium, and high). In addition, it assessed two building layouts in the model city:

- the "aligned" layout, which comprised 20 buildings located in a row, with the radial direction approximately parallel to the main axis of the valley (see Figure 2a);
- the "staggered" layout, which comprised 18 buildings located in a checker board layout (see Figure 2b).

Herein, the aligned and staggered building layouts in the original topographic configurations were implemented with low inflow discharge (see Figure 2c).

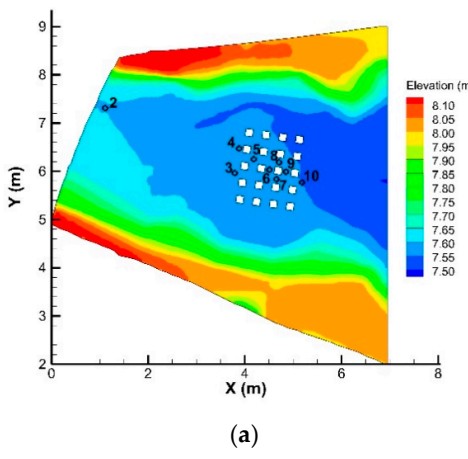 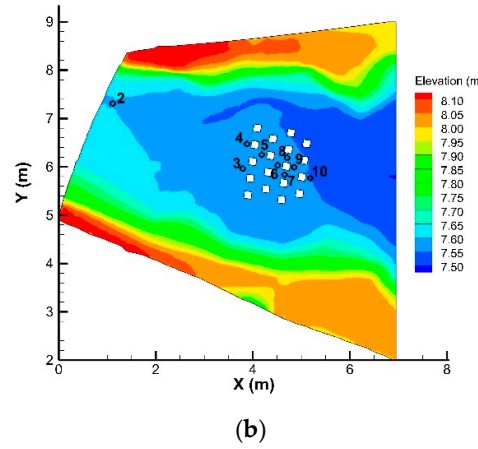

(**a**)                                             (**b**)

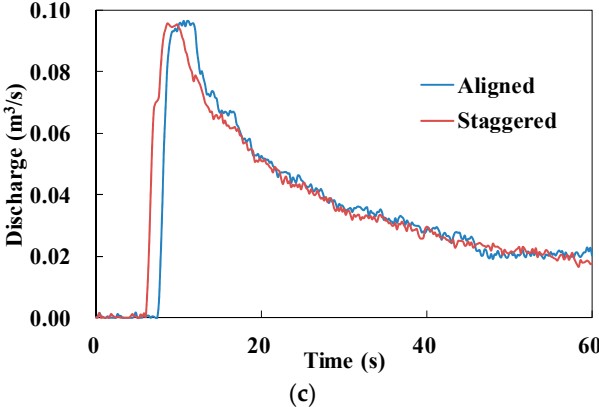

**Figure 2.** Toce River Valley case: (**a**) aligned building layout and gauge locations; (**b**) staggered building layout and gauge locations; (**c**) inflow discharge hydrograph.

### 2.3. Three Different Building Representations

Three building representation methods were considered to simulate the drag exerted by buildings, as follows:

#### 2.3.1. Building–Hole Method

The building–hole (BH) method assumes that the water flow would neither overflow nor permeate through buildings. The model of urban buildings is excluded from the simulation and the computational mesh is generated with holes aligned with buildings. This method sets up the holes with solid boundary conditions (see Figure 3). This approach ensures that no water flow occurs in the buildings.

#### 2.3.2. Building–Block Method

The building–block (BB) method increases ground elevation in the area of simulated buildings by modifying the distributed ground elevation data. By setting the buildings to a real height or an artificially high elevation value large enough to ensure that no water flows over the buildings, the entire simulated building area could be meshed as a unified grid, with no missing grid. In this method, the flood would flow around the buildings because the elevations of these simulated buildings are too high to allow overflow (see Figure 4). However, this method requires a localized refinement grid around buildings to accurately depict the building profiles.

#### 2.3.3. Building–Resistance Method

The building–resistance (BR) method assigns different Manning coefficients to each grid. The higher is the Manning coefficient, the lower is the water flow velocity. For the simulated building area, the Manning coefficient is set to a huge value to represent the resistance effect of the buildings. For other areas of simulated buildings, the Manning coefficient is set to a reasonable value to represent reality (see Figure 5). Therefore, this method can allow direct flood flow through the buildings, which is a simplification of reality.

### 2.4. Numerical Experiment Setup

Six numerical models were established for two building layouts (aligned and staggered) and three building representations (BH, BB, BR). To accurately simulate the terrain and profile of the buildings, a 2 cm grid resolution and 0.005 s time step were used. Grid resolution is an important parameter in numerical models [24,25]. For the TELEMAC-2D model, Hu indicated that the order of relative error monotonously decreases with increasing cell element number [26]. Jochen et al. stated that less convergence error occurs when a 2 cm resolution grid is used than that when a 1 cm

resolution grid is used, but the model ran an order of magnitude faster in the Glasgow case [27]. Therefore, in the Toce River Valley case, with a scale of 1:100 in the laboratory, the 2 cm resolution grid was adopted as a tradeoff between computational effort and model performance. The analysis of different mesh resolutions indicates the 2 cm resolution grid has great performance (see Section 3.1). To maximize accuracy, the time step should be controlled by the Courant number; thus, a 0.005 s time step has been applied in several model tests. The maximum Courant number value of the six numerical models used in [21] was 2.2 and the average Courant number was about 0.4. Using the same grid resolution and time step for each model helps mitigate their effects on the numerical results. Additionally, the topography was discretized by 94,629 nodes and 187,476 triangular elements using the BlueKenue free software. The solver accuracy of TELEMAC-2D is $10^{-8}$, which is sufficiently accurate to model the aforementioned scenarios.

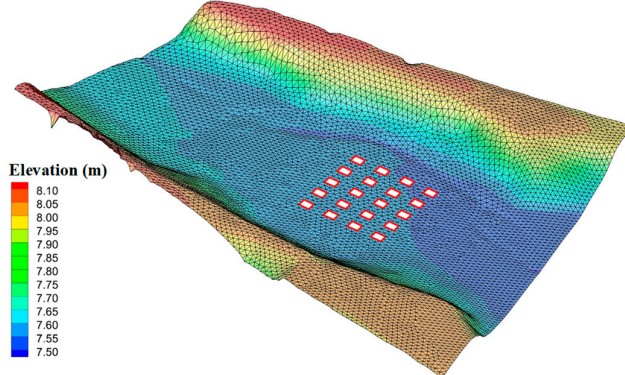

**Figure 3.** Numerical mesh of the building–hole (BH) method.

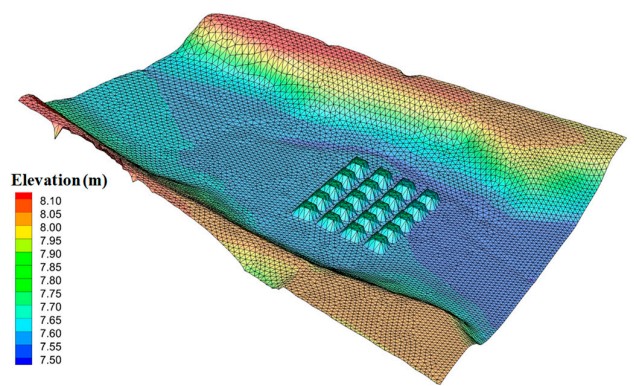

**Figure 4.** Numerical mesh of the building–block (BB) method.

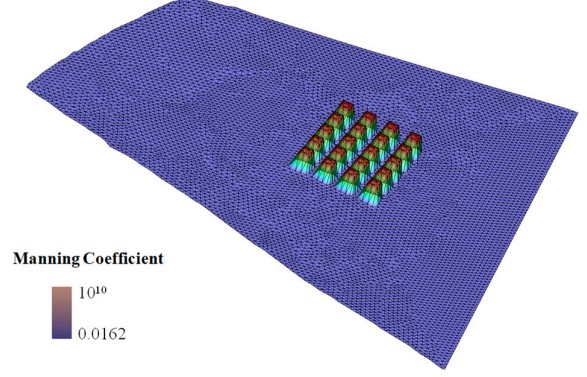

**Figure 5.** Numerical mesh of the building–resistance (BR) method.

In the present study, gauge P1 was not considered because it is located inside the upstream feeding tank. Gauge P2 is close to the flow entrance where the inflow is supercritical; therefore, the hydrograph elevation of the inflow was specified based on the measurements performed at P2. The hydrograph of the upstream discharge condition is presented in Figure 2c. The observed data measured by the other eight gauges at different locations are used for the purposes of analysis and comparison between measurements and predictions.

The 7th version of the TELEMAC-2D model was used for the numerical simulations based on FEM. In the BH method, the mesh was set up with several holes in the area of simulated buildings. In the BB method, the elevation of the building blocks was raised to 15 cm to emulate reality. In the BR method, the Manning coefficient was set to $10^4$ m$^{-1/3} \cdot$ s for building grids and 0.0162 for other grids, as reported by Testa [21]. Further information on the Manning coefficient for the building grid is presented in Section 3.2.

## 3. Results

### 3.1. Analysis of Different Mesh Resolutions

This section provides a detailed description and analysis of the water depth results for each mesh resolution.

To analyze this case more comprehensively, the effect of mesh resolution on the numerical model results is assessed, which may improve our understanding of numerical modeling in complex urban situations. The BH method (which uses a relatively simple modeling process) was used to analyze mesh resolution sensitivity at 1, 2, and 5 cm grid resolutions. Because of the different mesh resolutions, the time steps were also adjusted, and the average Courant number was set to about 0.4 to obtain convergent and accurate numerical results.

Results of the model predictions at the 1, 2, and 5 cm grid resolutions and laboratory measurements are shown in Figures 6 and 7. According to the graphs shown in Figures 6 and 7, the water depth results for the 5 cm grid model are larger than those for the other grid model resolutions, with the peak depth values showing the same trend. However, there are some exceptions, for example, the depth results at gauges P5 and P6 in the staggered layout with a 5 cm mesh resolution are lower than those of the other model results and have the lowest peak depth values. The depth results for the 1 cm grid resolution are slightly lower than those for the 2 cm grid resolution. Additionally, more obvious data oscillation can be observed in the 1 cm resolution results, particularly at gauges P9 and P10, which demonstrate partly that model performance for the 1 cm mesh resolution is more accurate.

Bennett et al. provided several qualitative and quantitative methods to validate model performance [28]. Of these, three indicators are chosen herein: root mean square error (RMSE), Pearson product–moment correlation coefficient (PPMCC), and Nash–Sutcliffe efficiency (NSE), which are classical and widely used quantitative indicators for assessing the accuracy of simulation results based on a comparison of predicted and measured values. Among these three indicators, RMSE is a frequently used measure to identify the differences between predictions and measurements, whereas PPMCC is used to evaluate the linear correlation between predictions and measurements, and NSE is used to assess the predictive power of hydrological models.

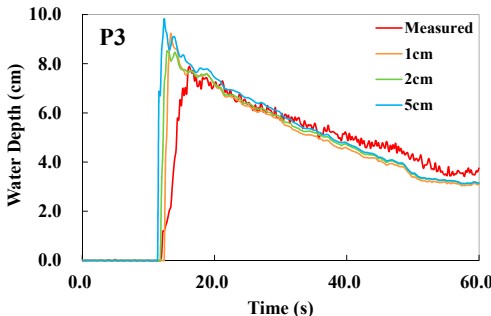
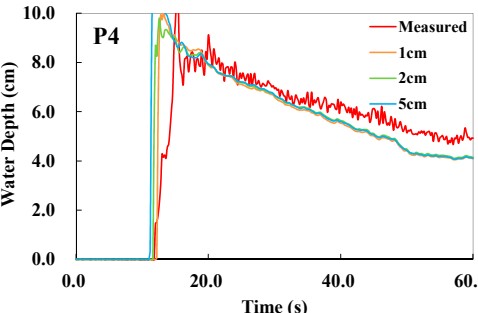

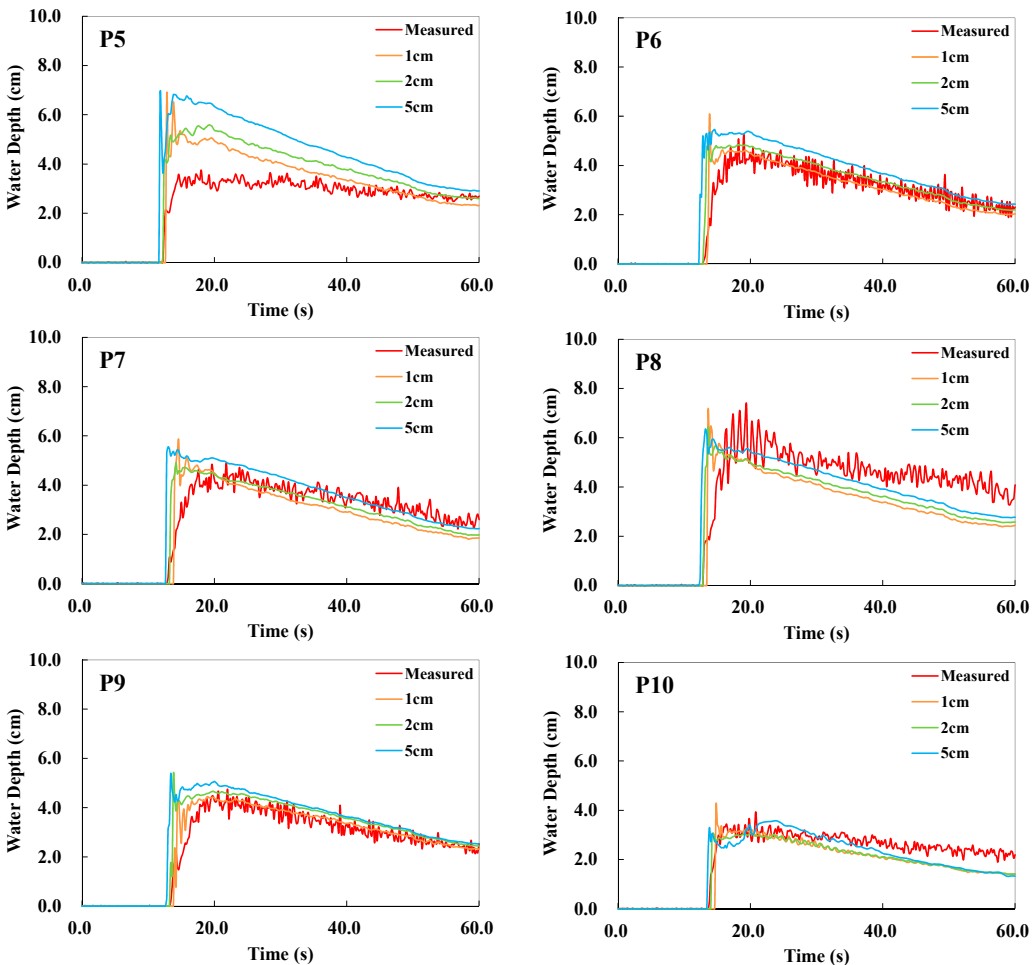

**Figure 6.** Effect of the mesh resolution of water depth in the aligned case.

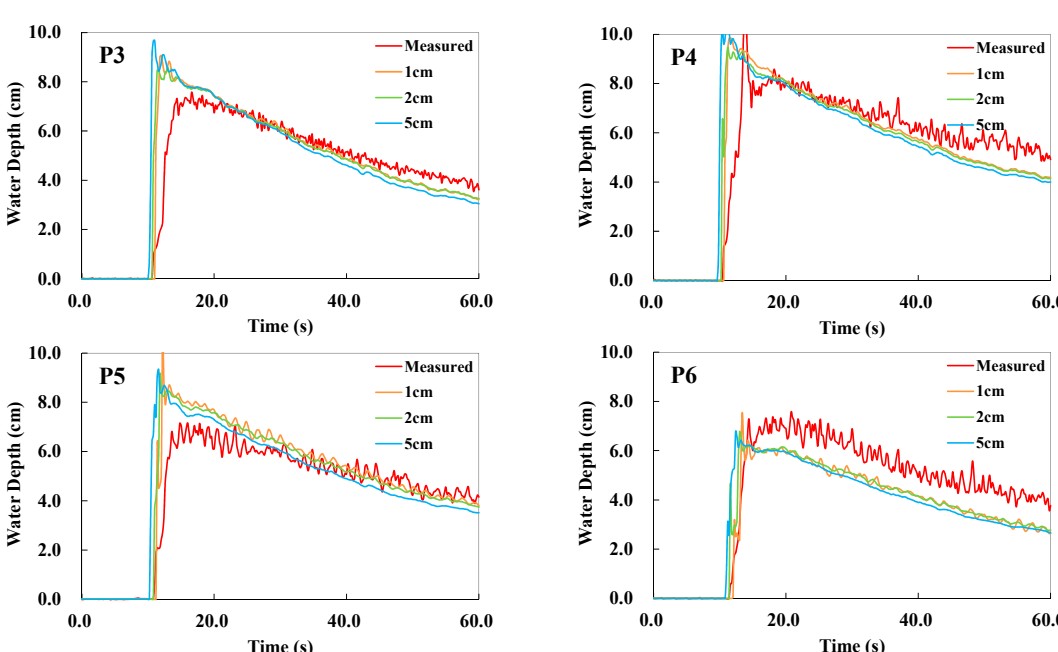

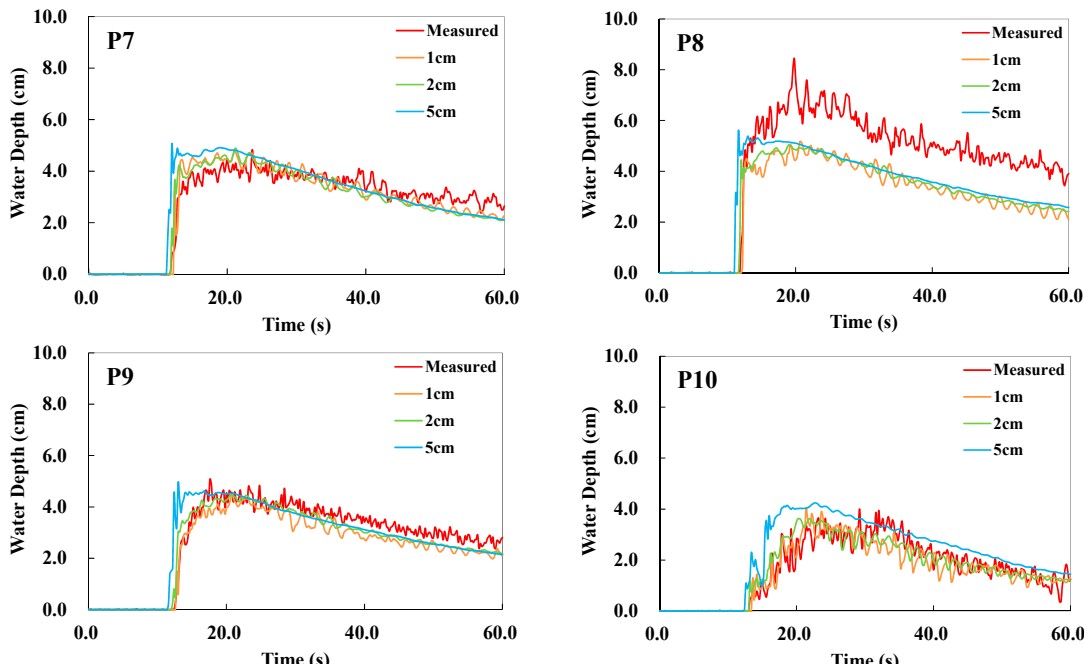

**Figure 7.** Effect of the mesh resolution of water depth in the staggered case.

In this section, we use RMSE, PPMCC, and NSE to quantify the performances of the different mesh resolutions.

We used the RMSE, PPMCC, and NSE indicators to quantify the performance of the three mesh resolutions. As shown in Table 1, the RMSEs of the 1 cm mesh resolution are lowest and those of the 5 cm mesh resolution are highest. In Tables 2 and 3, the PPMCCs and NSEs of the 1 cm mesh resolution are highest and those of the 5 cm resolution are lowest. The three indicators demonstrate that the 1 cm mesh resolution model shows the best performance. Based on the analysis of urban flood inundation extent, Yu et al. also reported that model performance is sensitive to mesh resolution [25]. Thus, based on Tables 1–3, the differences between the 1 cm mesh resolution and the 2 cm mesh resolution are small, and the model performance of 5 cm resolution is poor. Figures 6 and 7 and Tables 1–3 thus lead to the conclusion that finer mesh resolution provides better model performance. The model results are sensitive to mesh resolution.

**Table 1.** Root mean square errors (RMSEs) for the two building layouts and three mesh resolutions (/cm).

| Building Layouts | Mesh Resolutions | P3 | P4 | P5 | P6 | P7 | P8 | P9 | P10 | Average |
|---|---|---|---|---|---|---|---|---|---|---|
| | 1 cm | 1.03 | 1.19 | 0.90 | 0.48 | 0.68 | 1.24 | 0.40 | 0.55 | 0.81 |
| Aligned | 2 cm | 1.14 | 1.34 | 1.12 | 0.57 | 0.61 | 1.10 | 0.61 | 0.51 | 0.88 |
| | 5 cm | 1.55 | 1.83 | 1.84 | 0.96 | 0.92 | 1.06 | 0.84 | 0.53 | 1.19 |
| | 1 cm | 1.06 | 1.21 | 0.97 | 1.02 | 0.44 | 1.59 | 0.47 | 0.47 | 0.90 |
| Staggered | 2 cm | 1.14 | 1.32 | 1.02 | 1.06 | 0.47 | 1.49 | 0.38 | 0.44 | 0.92 |
| | 5 cm | 1.53 | 1.76 | 1.34 | 1.32 | 0.69 | 1.47 | 0.63 | 0.82 | 1.20 |

**Table 2.** Pearson product–moment correlation coefficients (PPMCCs) for the two building layouts and three mesh resolutions.

| Building Layouts | Mesh Resolutions | P3 | P4 | P5 | P6 | P7 | P8 | P9 | P10 | Average |
|---|---|---|---|---|---|---|---|---|---|---|
| | 1 cm | 0.84 | 0.83 | 0.83 | 0.91 | 0.84 | 0.79 | 0.94 | 0.89 | 0.86 |
| Aligned | 2 cm | 0.80 | 0.79 | 0.88 | 0.90 | 0.85 | 0.81 | 0.91 | 0.91 | 0.86 |
| | 5 cm | 0.67 | 0.63 | 0.77 | 0.83 | 0.76 | 0.77 | 0.85 | 0.84 | 0.76 |
| | 1 cm | 0.84 | 0.82 | 0.89 | 0.95 | 0.92 | 0.94 | 0.97 | 0.85 | 0.90 |
| Staggered | 2 cm | 0.81 | 0.78 | 0.86 | 0.91 | 0.91 | 0.93 | 0.95 | 0.86 | 0.88 |
| | 5 cm | 0.67 | 0.63 | 0.72 | 0.81 | 0.82 | 0.83 | 0.84 | 0.80 | 0.77 |

**Table 3.** Nash–Sutcliffe efficiencies (NSEs) for two building layouts and three mesh resolutions.

| Building Layouts | Mesh Resolutions | P3 | P4 | P5 | P6 | P7 | P8 | P9 | P10 | Average |
|---|---|---|---|---|---|---|---|---|---|---|
| | 1 cm | 0.82 | 0.82 | 0.47 | 0.90 | 0.81 | 0.64 | 0.93 | 0.78 | 0.77 |
| Aligned | 2 cm | 0.77 | 0.77 | 0.19 | 0.86 | 0.84 | 0.72 | 0.84 | 0.81 | 0.72 |
| | 5 cm | 0.58 | 0.57 | −1.18 | 0.60 | 0.64 | 0.74 | 0.68 | 0.80 | 0.43 |
| | 1 cm | 0.80 | 0.80 | 0.81 | 0.82 | 0.91 | 0.91 | 0.91 | 0.84 | 0.85 |
| Staggered | 2 cm | 0.77 | 0.76 | 0.79 | 0.80 | 0.90 | 0.58 | 0.94 | 0.86 | 0.80 |
| | 5 cm | 0.59 | 0.58 | 0.65 | 0.70 | 0.78 | 0.59 | 0.84 | 0.50 | 0.65 |

*3.2. Manning Coefficient Sensitivity Analysis*

The Manning coefficient is an empirical coefficient that estimates the average flow velocity, which crucially affects numerical models [29]. In the BR method, several similar numerical models were established using different Manning coefficients for building grids to assess the coefficient's sensitivity. The indicators used for model performance were unit-width discharge in the building grids and water depth, which could appropriately reflect the resistance effect of buildings (see Figure 8). The unit-width discharge in the building grids indicates the amount of flood flow through the building grids across a unit-width. The water depth is the depth recorded at a point with an x-coordinate of 4.65 and y-coordinate of 6.02 (located in the building grids). The Manning coefficients were $10^0$–$10^{10}$ $m^{-1/3} \cdot s$. Figure 8 shows the results: when the Manning coefficient is less than $10^4$ $m^{-1/3} \cdot s$, the unit-width discharge and water depth in the building grids decreases as the Manning coefficient increases. In contrast, when the Manning coefficient is $10^4$–$10^{10}$ $m^{-1/3} \cdot s$, the unit-width discharge and water depth in the building grids have almost no discharge, and the trend of the unit-width discharge and water depth in building grids has no change. Therefore, the unit-width discharge and water depth in the building grids are only slightly sensitive when the Manning coefficient is greater than $10^4$ $m^{-1/3} \cdot s$. However, even when the Manning coefficient was assigned a very high value, model performance was not significantly improved. Therefore the Manning coefficient for building grids is set to $10^4$ $m^{-1/3} \cdot s$ in the BR method.

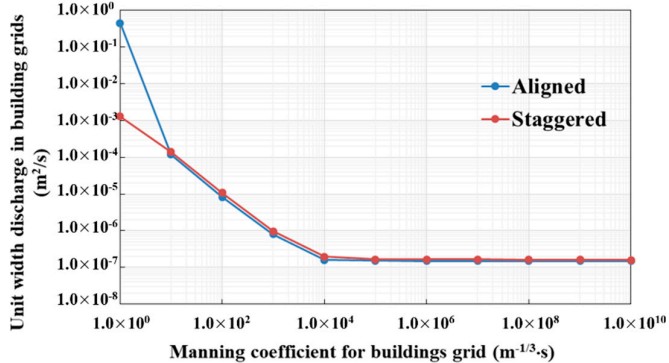

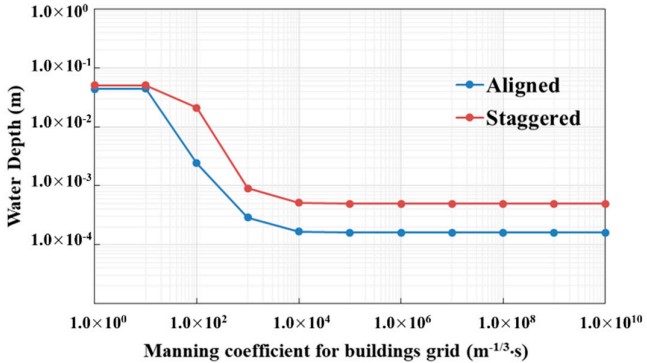

**Figure 8.** Manning coefficient sensitivity.

### 3.3. Analysis of Results for Different Building Representations

This section provides a detailed description and analysis of the water depth results for different building representations.

Figure 9 presents the results of model predictions and laboratory measurements for the aligned case at gauges P3–P10. Some obvious discrepancies can be observed at P5 compared with the other seven gauges. The measured values are lower than the predicted values for gauge P5, with the models for the three building representation methods yielding the same results. However, the other seven gauges show good performance, with the simulation water levels being consistent with the laboratory values. When comparing the predictions with the measurements, clearly, the variation in water level matches, with the time peak water-level value showing only minor differences. In addition, further analysis of the differences between the predictions and measurements clearly reveals that the simulation water levels follow the order BB > BR > BH (except at gauge P10, where the simulation water level for BR is highest and that for BB is intermediate).

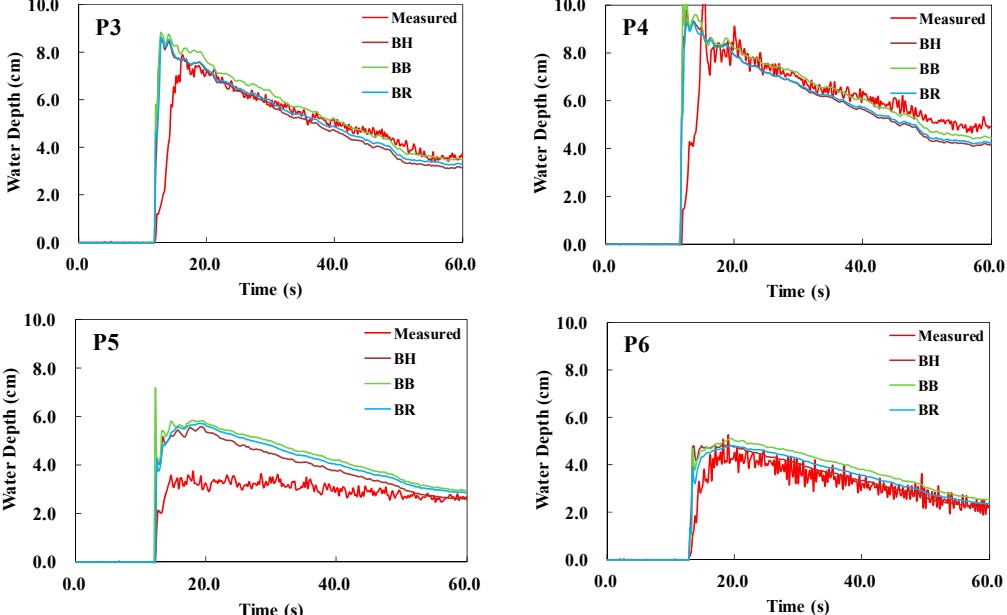

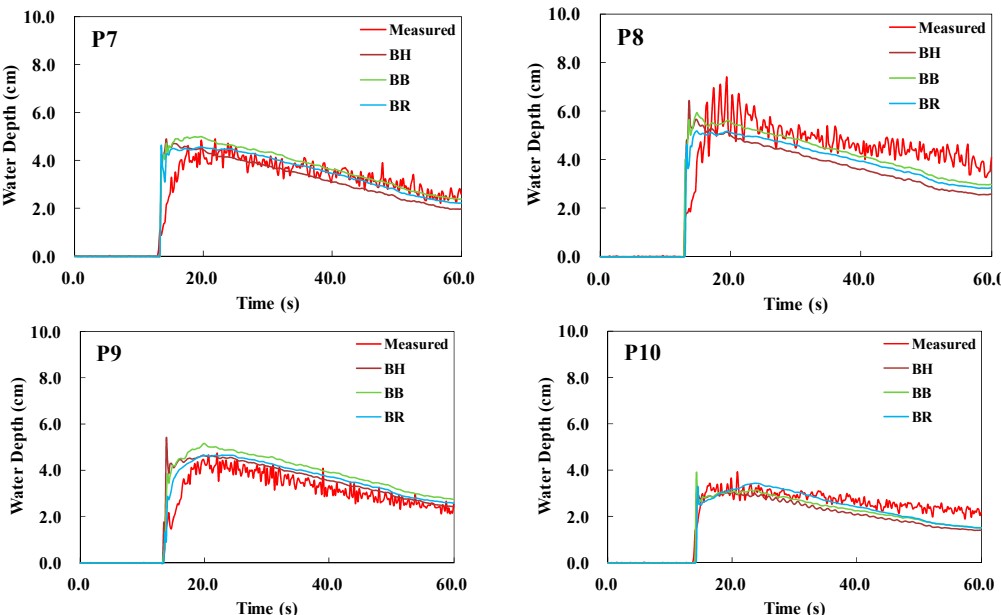

**Figure 9.** Prediction and measurement results for water depth in the aligned case.

Figure 10 presents the model prediction and laboratory measurement results for the staggered case at gauges P3–P10. The simulation water levels for the different building representation models are in accordance with the laboratory values. With the staggered building layout, the predicted values for gauge P5 match the laboratory values more closely than they do for the aligned building layout. Although the values at gauge P8 show an obvious difference, the predictions are lower than the measurements for gauge P8. Gauge P6 exhibits the same performance as gauge P8, but the differences are not as marked as they are for P8. On comparing the three model values, the staggered layout exhibited similarities with the aligned layout; i.e., the simulation water level for the BB method, is highest and that of the BH method is lowest.

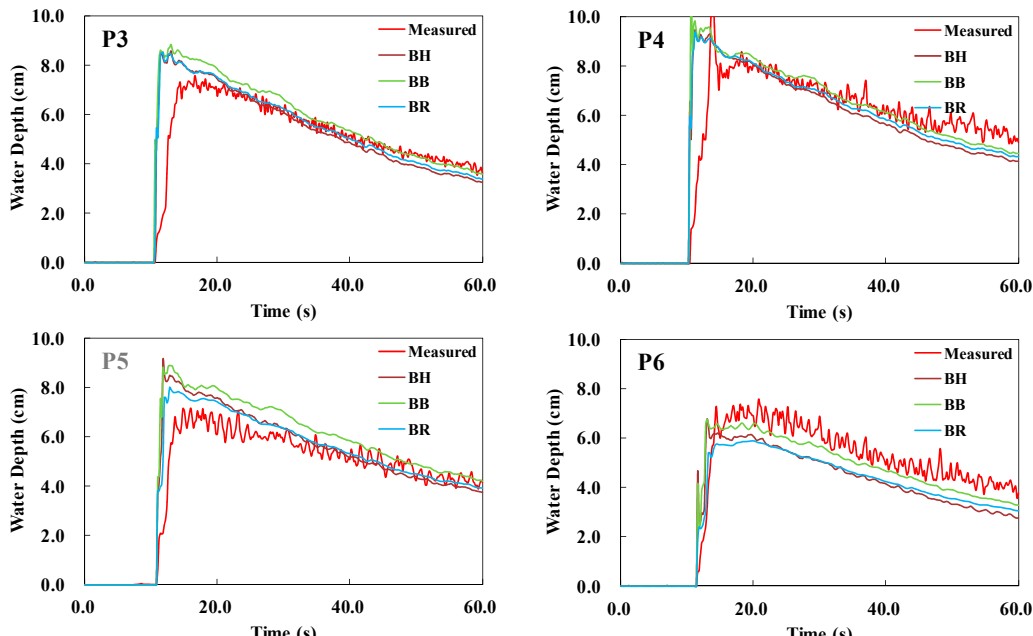

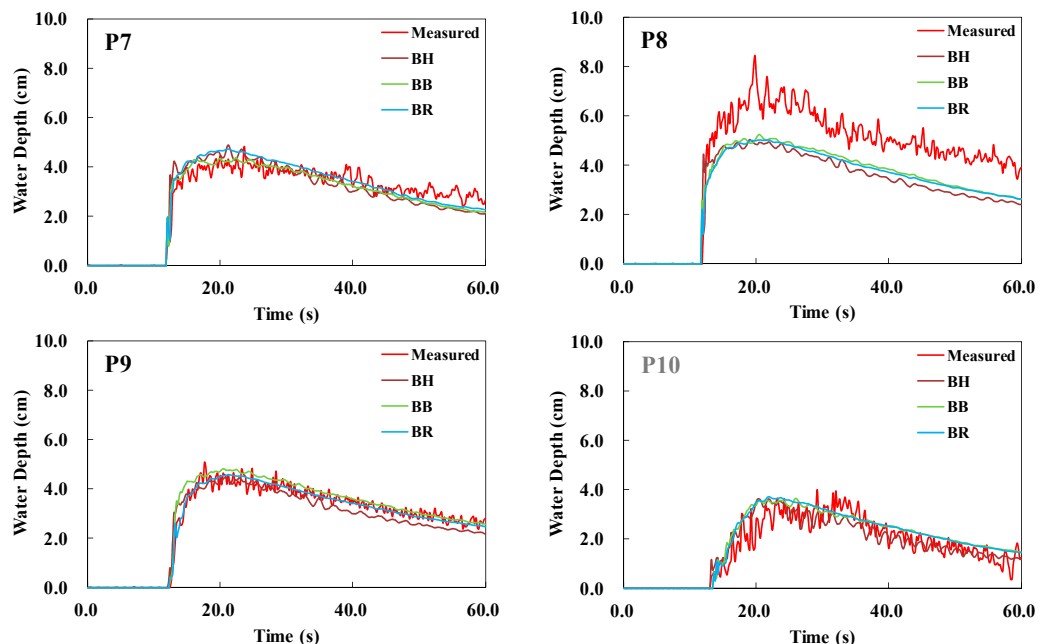

**Figure 10.** Predicted and measured water depths in the staggered case.

Results indicate that the TELEMAC-2D model can accurately simulate flood flow in this case. Comparing the predictions made herein with those made in other previous studies that adopted the same case [15,30] reveals that the simulations performances of previous studies at gauges P5 and P8 were also relatively poor. Thus, these findings suggest that TELEMAC-2D is applicable to urban flood simulations.

In this section, similarly, we use RMSE, PPMCC, and NSE to quantify the performances of the aforementioned three methods.

Table 4 shows that the average RMSEs of the three methods have some obvious discrepancies in the aligned case: the value of the BB method is highest and that of the BR method is lowest. In the staggered case, the value of the BH method is highest and that of the BR method is lowest. In general, the average RMSEs of the three methods are between 0.80 and 0.92, indicating that some differences exist between the predictions and measurements.

**Table 4.** Root mean square errors (RMSEs) for two building layouts and three building representations (/cm).

| Building Layouts | Building Representations | P3 | P4 | P5 | P6 | P7 | P8 | P9 | P10 | Average |
|---|---|---|---|---|---|---|---|---|---|---|
| | BH method | 1.14 | 1.34 | 1.12 | 0.57 | 0.61 | 1.10 | 0.61 | 0.51 | 0.88 |
| Aligned | BB method | 1.23 | 1.37 | 1.48 | 0.70 | 0.60 | 0.81 | 0.74 | 0.42 | 0.92 |
| | BR method | 1.11 | 1.24 | 1.29 | 0.51 | 0.51 | 0.85 | 0.51 | 0.39 | 0.80 |
| | BH method | 1.14 | 1.32 | 1.02 | 1.06 | 0.47 | 1.49 | 0.38 | 0.44 | 0.92 |
| Staggered | BB method | 1.26 | 1.40 | 1.25 | 0.74 | 0.35 | 1.27 | 0.31 | 0.50 | 0.89 |
| | BR method | 1.11 | 1.26 | 0.80 | 0.93 | 0.38 | 1.33 | 0.20 | 0.49 | 0.81 |

Table 5 shows that the average PPMCCs of the three methods show the same trend for the aligned and staggered cases: the value of the BR method is highest and that of the BH method is lowest. The average PPMCCs of the three methods are all greater than 0.86, indicating that the predictions and measurements have a strong linear correlation.

**Table 5.** Pearson product–moment correlation coefficients (PPMCCs) for the two building layouts and three building representations.

| Building Layouts | Building Representations | P3 | P4 | P5 | P6 | P7 | P8 | P9 | P10 | Average |
|---|---|---|---|---|---|---|---|---|---|---|
| Aligned | BH method | 0.80 | 0.79 | 0.88 | 0.90 | 0.85 | 0.81 | 0.91 | 0.91 | 0.86 |
| | BB method | 0.81 | 0.79 | 0.86 | 0.94 | 0.91 | 0.86 | 0.95 | 0.92 | 0.88 |
| | BR method | 0.81 | 0.81 | 0.90 | 0.94 | 0.91 | 0.89 | 0.96 | 0.91 | 0.89 |
| Staggered | BH method | 0.81 | 0.78 | 0.86 | 0.91 | 0.91 | 0.93 | 0.95 | 0.86 | 0.88 |
| | BB method | 0.81 | 0.77 | 0.88 | 0.93 | 0.95 | 0.96 | 0.98 | 0.87 | 0.89 |
| | BR method | 0.82 | 0.80 | 0.92 | 0.97 | 0.95 | 0.97 | 0.98 | 0.88 | 0.91 |

Table 6 show that the average NSEs of the three methods exhibit different trends in the aligned case: the value of the BR method is highest and that of the BB method is lowest. However, in the staggered case, the value of the method BR is highest and that of the BH method is lowest. Table 6 also shows that the NSEs of gauge P5 in the aligned case and those of gauge P8 in the staggered case are lower than the average NSEs; their poor performances are shown in Figures 9 and 10. In general, the average NSEs of all scenarios are greater than 0.65, indicating that TELEMAC-2D with the three building representations is sufficiently accurate for urban flood simulations.

Tables 5 and 6 show that the average PPMCCs and NSEs in the aligned case are lower than those in the staggered case, but for the average RMSEs in Table 4, the trend is unclear. Although some differences exist for each of the indicators at every gauge, TELEMAC-2D performs much better in the staggered case than in the aligned case. Furthermore, the BR method shows better performance by comparing the three indictors, it yields relatively low RMSEs and high PPMCCs and NSEs.

**Table 6.** Nash–Sutcliffe efficiencies (NSEs) for the two building layouts and three building representations.

| Building Layouts | Building Representations | P3 | P4 | P5 | P6 | P7 | P8 | P9 | P10 | Average |
|---|---|---|---|---|---|---|---|---|---|---|
| Aligned | BH method | 0.77 | 0.77 | 0.19 | 0.86 | 0.84 | 0.72 | 0.84 | 0.81 | 0.72 |
| | BB method | 0.74 | 0.76 | -0.41 | 0.79 | 0.85 | 0.85 | 0.75 | 0.87 | 0.65 |
| | BR method | 0.79 | 0.80 | -0.07 | 0.89 | 0.89 | 0.83 | 0.88 | 0.89 | 0.74 |
| Staggered | BH method | 0.77 | 0.76 | 0.79 | 0.80 | 0.90 | 0.58 | 0.94 | 0.86 | 0.80 |
| | BB method | 0.72 | 0.74 | 0.69 | 0.90 | 0.94 | 0.69 | 0.96 | 0.82 | 0.81 |
| | BR method | 0.78 | 0.78 | 0.88 | 0.85 | 0.93 | 0.66 | 0.98 | 0.83 | 0.84 |

## 4. Discussion

### 4.1. Model Performance with Different Building Representations, Manning Coefficients, and Mesh Resolutions

The Manning coefficient is an extremely significant parameter in numerical models with the BR method. When the Manning coefficient is greater than $10^4$ $m^{-1/3} \cdot s$, numerical model performance does not improve, which is of great practical importance in such models. While establishing an urban flood numerical model, collecting accurate terrain and other urban data is difficult. Therefore, the resistance effect of buildings is expressed by the Manning coefficient. The conclusions of this study provide further support for the use of the Manning coefficient. However, the $10^4$ $m^{-1/3} \cdot s$ of Manning coefficient has no physical meaning in the real world, and is only useful in the numerical model. Because there are windows, basement, square and so on in building area, the flood could in fact flow through the building area partly. Therefore, the set of Manning coefficient not only depends on the collection of basic hydrologic data, but also depends on the calibration and validation of the numerical model, which should be agree with the actual states.

In terms of model performance, the BB, BH, and BR methods yielded results that were similar to the measurements, as shown in the hydrographs in Figures 9 and 10. However, the water depth predictions form more uniform and smoother curves than the measurements. Dottori et al. also reported this phenomenon [31]. An interesting observation in the hydrographs is that the BH

method exhibits greater data oscillation than the other two methods, indicating that the BH method has better performance in capturing the complex interaction between buildings and water flow than the other two methods. Additionally, for nearly all hydrographs, peak depth occurs a few seconds earlier in the predictions than it does in the measurements. This may be due to flow front celerity being influenced by mesh resolution; as observed in Figures 6 and 7, the peak time for 1 cm mesh resolution is a little behind that for 5 cm mesh resolution, which is closer to the measurements. Furthermore, the time spent for water to move differs from reality when the grids located in front of the buildings are changed from dry to wet [25]. An et al. also reported this phenomenon, citing the following potential reason: as the inflow rate of the experiment was calculated by gauging the tank upstream of the flow domain, a time gap occurred between the simulated and numerical models [16].

Mesh resolution influences model performance. The runtime computational cost of the 1, 2, and 5 cm mesh resolutions are 2.5 h, 0.6 h, and 0.2 h respectively, which means that the finer is the mesh resolution, the greater is the central processing unit (CPU) runtime. The water depth peak value and peak time also showed differences across the mesh resolutions. For certain gauges, e.g., P3, P4, and P5, which were stroke by flow directly, the predicted peak depth values are greater than the measured values at the different mesh resolutions. For other gauges, e.g., P7, P9, and P10, which were located at the back of the building zones, the predicted peak depth values are slightly smaller than the measured values at different mesh resolutions. Although different mesh resolutions were implemented, the phenomenon of the predicted time of water depth peak being ahead of the measurements did not change, as shown in Figures 6 and 7. However, the peak depth time for 1 cm mesh resolution showed slight improvement compared with the results of the 5 cm mesh resolution, but this is not very subtle. Data oscillation is more apparent for the results of 1 cm mesh resolution compared with that of 2 and 5 cm mesh resolutions. This oscillation is more noticeable for the staggered layout than the aligned layout, particularly in gauges P6, P8, and P9. This observation indicates that reducing the mesh resolution may help the numerical model capture data oscillation.

The aforementioned phenomenon should be noted and considered in numerical models in the future.

### 4.2. Abnormal Results Analysis

Explaining the results of gauge P5 in the aligned case and those of gauge P8 in the staggered case is difficult. These results may be related to the potential for complex flow fields at these two locations, which might cause unusual water flow. Some possible explanations are as follows.

For gauge P5 in the aligned case, the flood passed through the middle of two buildings in the first row; therefore, the P5 location was impacted by water shock at about 12.4 s, before the water jump occurred. An important finding was that the simulated values of the water jump's depth were greater than the observed values. Similar water jump values can be found at gauges P3 and P4 in the aligned and staggered cases, respectively. After several seconds, the simulated values are consistent with the measurements (see Figures 9 and 10). For example, after ~20 s, the simulated results of gauges P3 and P4 are very accurate, and after about 50 s, the simulated result of gauge P5 also becomes accurate. This phenomenon indicates that numerical models have some issues in dealing with abruptly changing values. Additionally, for gauge P5, which was located in the middle of four buildings and also faced the strike of the flood, the numerical model has some difficulties in simulating the oscillation of the values in the complex flow field. However, the discrepancies in water depth variation between the predictions and measurements were corrected over time; eventually, they will be eliminated.

Because P7 and P8 were located in symmetrical positions in the staggered case, comparing the results obtained at these two locations reveals that the predicted values for two sites are similar; however, the measurement values are obviously different. A possible explanation for this is that a concealed channel groove exists in the terrain to the north of the buildings (see Figure 2). Figure 11 shows that water velocity is faster in the channel groove. Therefore, the channel groove has some influence on velocity and depth at gauge P8 (see Figure 11). According to Bernoulli's equation,

higher flow speed occurs at the point where pressure is lower. Therefore, dragging due to the high-speed flow in the channel groove causes more water to flood gauge P8 in the northeast direction; i.e., greater water flow occurs at gauge P8 and the water depth therein is slightly higher than that at gauge P7. In addition, Figure 11 shows that the channel groove has a greater influence in laboratory measurements, resulting in a higher water depth at gauge P8. The hydrographs of predicted and measured results at gauge P8 are almost parallel after 30 s (Figure 10), suggesting that the influence of the channel groove is underestimated in the numerical model when compared with reality.

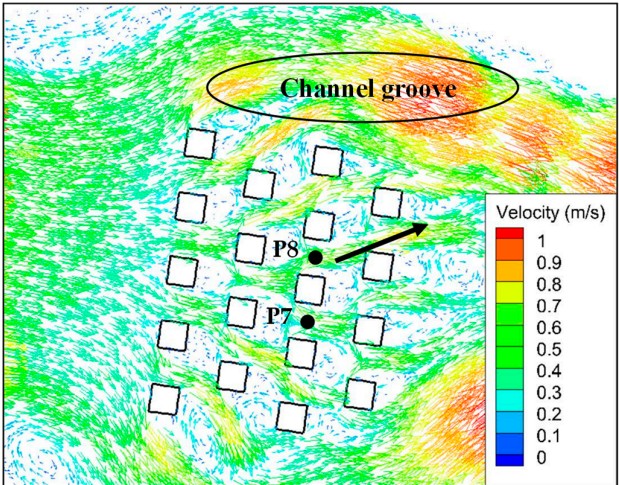

**Figure 11.** Flow velocity field at 20 s in the staggered case.

The results of the Toce River Valley case demonstrate that certain challenges exist when using numerical models to simulate resistance under complex flow conditions. These results might be explained in part by the assumptions of classical SWEs, e.g., the pressure distribution is approximately hydrostatic and the vertical velocity is ignored. Owing to the existence of the water jump, the channel groove, and other complex flow conditions, the limitations of the SWEs must be recognized [32]. Thus, further research is required to deal with complex flow.

The simulated velocity profiles were analyzed using the three building representation methods and the two building layouts. We present the velocity profiles along the line segment between gauges P5 and P8 at 15 s in Figure 12. In the aligned layout, no building existed on this line; however, two buildings existed on this line in the staggered layout. Interestingly, the velocity of the BB method is lowest among the three methods at gauge P5 in the aligned case and at gauge P8 in the staggered case. Figures 9 and 10 show that the water depth is greater at the P5 and P8 locations, partially confirming the law of conservation of mass. In addition, for the BB method, the flow velocities are not zero in building zones, indicating that the water overflowed the building zones (although the velocities are relatively small). These results demonstrate that in contrast to the BH and BR methods, the BB method could not prevent the water from entering the building zone. In order to simulate the urban buildings realistically, setting an artificially high elevation value could prevent the water from entering the building zone in the BB method and future studies on this phenomenon are recommended. Based on the velocities along the line segment, some differences exist among the three building representation methods. Owing to the absence of observed velocity data, the simulated velocities could not be assessed in relation to the measurements.

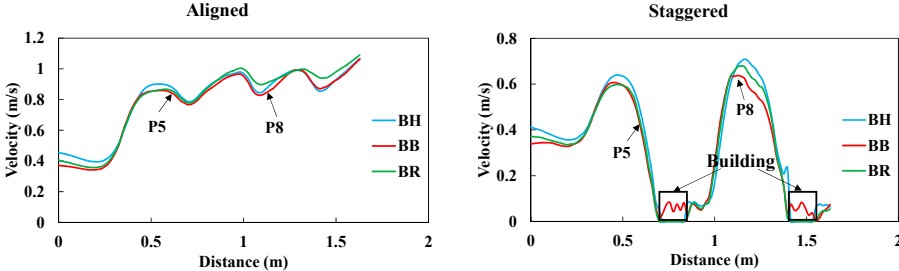

**Figure 12.** Simulated velocity profiles at 15 s.

The Froude number is used herein to further analyze the interactions between buildings and water flow.

Maps of the Froude number were drawn for an instant of time (at about 15 s) at the peak water depth with three building representation methods and two layouts. As shown in Figure 13, there are complex flow regimes around the buildings. Because of the difference between the inflow discharges in two layouts (see Figure 2c), we could find the flood flow of aligned layout has a lag time of 1.4 s to the staggered layout. In the same building layout, it is clear that the maps of Froude number are pretty similar, such as Figure 13a–c or Figure 13d–f, and the flow regimes around the building area are same under the different building representation methods, especially the interactions of subcritical and supercritical flows. Behind the buildings and along the flow direction, there is subcritical flow, whereas in the middle of the buildings and along the flow direction, there are subcritical and supercritical flows around the buildings. When the flow field is stable after 15 s, the distributions of subcritical and supercritical flows are also stable. Furthermore, TELEMAC-2D provides good performance during the transition between subcritical and supercritical flows. Additionally, the results of Froude number maps demonstrate once again that TELEMAC-2D with the three building representations is sufficiently accurate for urban flood simulations.

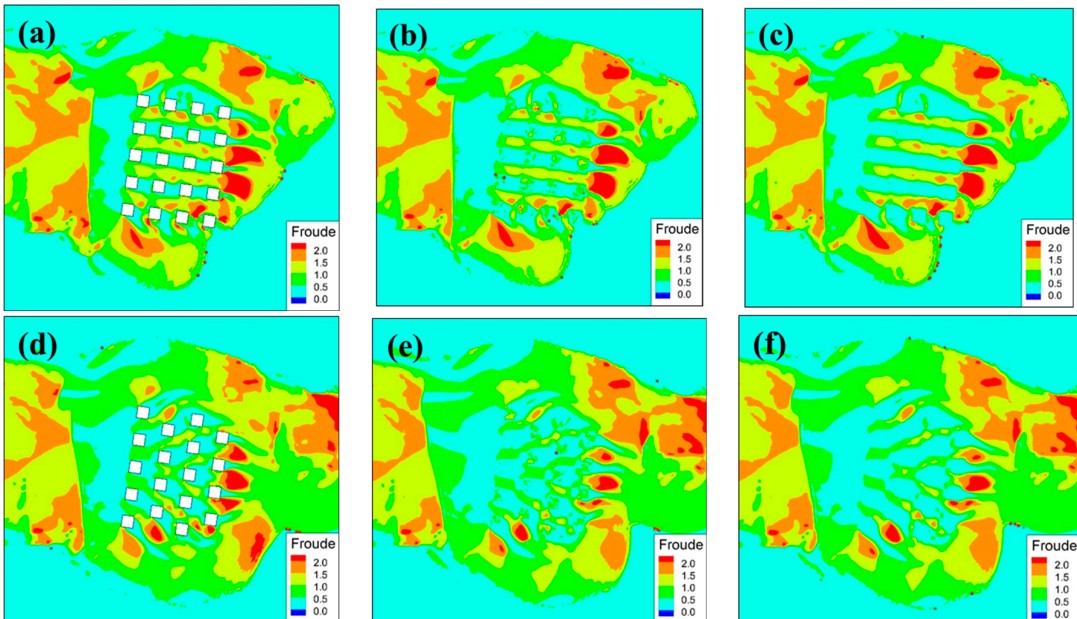

**Figure 13.** Froude number after 15 s with aligned and staggered layouts: (**a**,**d**) are in BH method; (**b**,**e**) are in BB method; (**c**,**f**) are in BR method; (**a**–**c**) are in aligned layout; (**d**–**f**) are in staggered layout.

Figure 14 shows the Froude numbers at gauges P5 and P8 in the aligned and staggered cases, respectively. Notably, the Froude numbers range mostly between 0.8 and 1.1 when the flow field

attains stability after 15 s. Critical flow has a complex flow regime and strong sensitivity toward water depth [33]. Thus, accurately simulating water flow depth is difficult when the modeled flow regime is critical.

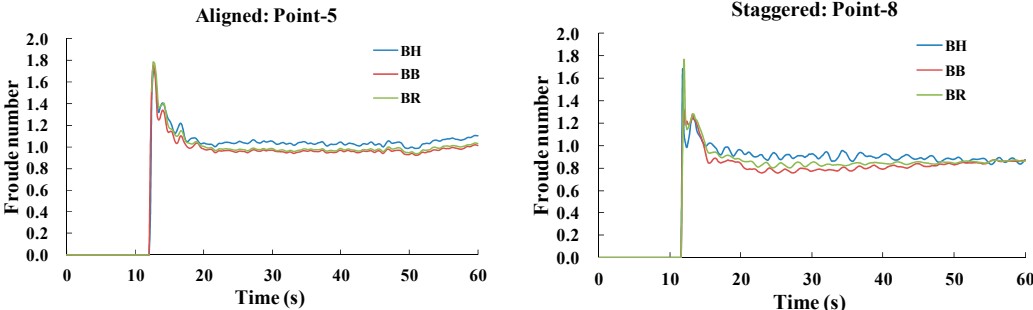

**Figure 14.** Froude number at gauges P5 and P8 in the aligned and staggered cases, respectively.

The different mesh resolutions have some influence on the two abnormal results outlined above, as shown in Figures 6 and 7. For gauge P5 in the aligned layout, the finer mesh resolution may help reduce the discrepancy between the predictions and measurements. For gauge P8 in the staggered layout, the finer mesh resolution has greater capacity to capture the data oscillation and thereby reflect reality. A finer grid resolution is better than a coarse resolution in terms of the depth value and peak time. However, a coarse grid with a classical SWE model would somewhat neglect water storage reduction and flow energy loss [34]. Therefore, mesh resolution has some sensitivity toward water depth value.

### 4.3. Further Considerations and Suggestions

This paper used the TELEMAC-2D model to analyze urban flooding by taking a classical experiment test Toce case and compared with previous researches. This paper has provided a great contribution to the problem related to urban flooding model benchmarking, focusing on the effect of different building representation. However, this study does not study detailed real-world events with numerical model, because the research of this paper is basic, and concentrates on the performance and accuracy of this model, and adopts different accuracy indicators to evaluate the simulated results. This study provides some support for using the TELEMAC-2D model to simulate real-world situations. There are many researches focus on the real-world urban flood. Segura-Beltran et al. studied a detailed case using surveys and geomorphologic map to evaluate the performance of the numerical model [35]. Macchione et al. adopted non-conventional information (such as amateur videos, photographs, news reports, etc.) to establish a past urban flood events model [36]. Martin-Vide et al. analyzed an extreme flash flood in Spain [37]. However, Neal et al. mentioned that it is difficult to assess the accuracy of the numerical simulation due to the lack of detailed inundation extent or water surface elevations [38]. Although there many difficulties to simulate a real-world urban flood, it is vital importance for urban management and development, also it points out the next research direction for the future.

In addition, more consideration can be given to the urban flood in the numerical model. Flood risk management is an important future research field. To this end, numerical modeling is a significant method owing to its efficiency and precision. In general, land-use maps showing impermeable surfaces, particularly gray infrastructure, in urban areas could be collected via refined aerial remote sensing. In urban flood models, some gray infrastructure cannot be infiltrated but water can pass through these surfaces; e.g., roads. Other infrastructure such as buildings cannot be simultaneously infiltrated and overflowed. Therefore, further details, particularly relating to topography and the height of gray infrastructure, need to be considered in actual urban flood models. The Manning coefficient is also highly significant in relation to roads because it represents

the flood velocity. In the context of flood risk prediction and damage assessment, further research into model buildings and setup-related parameters in urban districts is warranted.

Finally, as reported previously [39,40], several advanced techniques such as 3D flood hazard visualization have been implemented to better present numerical results. Numerical model results could provide support to urban flood management. The TELEMAC-2D model can simulate urban floods, as shown in this study, and as it is an open source code model, it is flexible and adaptive, which is advantageous to developing an urban flood model for risk management. Consequently, related urban flood research can be developed in the future, the results of which could be helpful in terms of flood management.

## 5. Conclusions

Urban buildings significantly impact urban flooding, particularly in terms of water depth, which is a crucial parameter for flood risk management and damage assessment. In this study, three building representation methods were tested in aligned and staggered building representations and a series of numerical models were established for analysis and comparison. Detailed water-level evolution processes around buildings were studied by comparing predicted and measured values. The main conclusions are as follows:

- In the BR method, the Manning coefficient is an important parameter because it represents the amount of flood flow passing through the building grids. Based on sensitivity analysis, the Manning coefficient is not sensitive when it is larger than $10^4$ m$^{-1/3} \cdot$ s; furthermore, the unit-width discharge and water depth in the building grids decreases as the Manning coefficient increases when the Manning coefficient is less than $10^4$ m$^{-1/3} \cdot$ s.

- The TELEMAC-2D model can simulate urban flooding with complex underlying surfaces. The model results obtained herein from the three building representation methods are mostly consistent with measured values. However, some differences exist in the simulation results: e.g., the peak water-level values are slightly greater than the measured values, and the peak time falls ahead of the observed value. Results demonstrate that the TELEMAC-2D model is largely applicable to the simulation of urban flooding. As there are obvious differences at some locations (e.g., at gauges P5 and P8), the modeling methods need to be further optimized and model parameters need to be reasonably adjusted based on actual complex situations.

- On comparing the three building representation methods, BB, BH, and BR, the three accuracy indicators, namely, RMSE, PPMCC, and NSE, clearly show that all three methods exhibit significant potential to simulate urban flooding with aligned or staggered building layouts. There is no appropriate method that would offer maximum performance in terms of all three indicators. Analysis of assessment indicators for different layouts of the model city revealed few differences between the three methods. In general, the BR method shows relatively better performance according to the three indicators. Hence, the BR method is deemed suitable for urban flood simulation.

- Numerical models with 1, 2, and 5 cm mesh resolutions for the BH method were established to analyze the influence of the sensitivity of different mesh resolutions. By comparing model performance, we found that the results for the 1 cm mesh resolution are slightly lower than those for the other mesh resolutions. In addition, data oscillation is visible in the results for the 1 cm resolution. Furthermore, RMSEs, PPMCCs, and NSEs were calculated to appraise quantitatively the performance of different mesh resolutions. The results for the 1 cm mesh resolution were found to be better that those for the other mesh resolutions based on the calculated indicators.

- The discrepancies in the predictions and measurements at gauges P5 and P8 were analyzed and discussed from different perspectives. A possible explanation for these discrepancies is that the water level was influenced by water jump and micro-topography, implying that the classical SWEs have some issues in terms of their hydrostatic and zero vertical velocity hypotheses. Additionally, the mesh resolution, velocity, and Froude number have some influence on the numerical results, and some difficulties in simulating critical flow were

experienced. Therefore, modifying and perfecting classical SWEs is an important research objective for future studies.

**Author Contributions:** Z.L. developed the numerical model and wrote the original manuscript. J.L. and C.M. analyzed the results and revised the manuscript. W.S. checked the datasets and performed the simulation. H.W. provided theoretical guidance. D.Y. corrected the use of English in the manuscript.

**Funding:** This study was supported by the National Key Research and Development Program of China (2016YFC0401401), the Chinese National Natural Science Foundation (No. 51739011 & No. 51879274), and the Research Fund of the State Key Laboratory of Simulation and Regulation of Water Cycle in River Basin (No.2017ZY02).

**Conflicts of Interest:** The authors declare no conflict of interest.

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
