# Peer review of "Comparative Analysis of Building Representations in TELEMAC-2D for Flood Inundation in Idealized Urban Districts"

_water, doi:10.3390/w11091840_

Round 1

Reviewer 1 Report

This is my third review of this manuscript.

In my view, the authors made several efforts to improve their paper, trying to take into account all the reviewers' comments.

Compared to the previous versions of the work, the present form of the manuscript is more clear and the main ideas are better discussed. 

However, though I've appreciated the revised version of the manuscript, I think that it still suffers from some drawbacks. 

In particular, some parts of the paper should be improved focusing better on what is the real interest of this work.

For example, I suggest:

1) to significantly reduce the first part of the introduction or to completely remove lines 1-47, because it contains general and too vague information.

2) to focus more on urban flooding and, specifically, to the experimental and real-world applications discussed in the literature. The authors could refer to two recent papers [1 and 2] in which they can find a lot of references related to experimental and real-world applications, respectively. So I think that these works, and related references, should at least mentioned in order to further appreciate the work done by the authors and the importance of their research in respect to the existing state of the art;

3) to motivate better the choice of the Toce test case. The authors have cited the work presented by Mignot et al. 2019. In that paper, the authors presented a lot of experimental tests of idealized urban flooding. Why did the authors considered only the Toce test case?

4) to provide some comments or justifications for which the authors decided to analyze only experimental test case and not detailed real-world events, since some cases of urban floods are available in the literature (see reference [2,3 etc]);

5) to take care at the Manning coefficient. It is not a dimensionless parameter. Please, add the related dimensions in the text and in the figures.

6) to discuss better  the suggested value of the Manning coefficient, since it is an extraordinary high value and probably it has no physical meaning. It should be considered as a calibration parameter. More comments are required on this. Finally, what kind of suggestion the authors give for practical application?

7) to analyze better the Froude number behavior. I suggest to organize a figure in which the authors could show the Froude number map throughout the domain, in a specific instant of time, not only for both the tests but also for each of three methods for taking into account the idealized city. In this way, the authors could provide original comments on the effects of the three techniques on the flow behavior, in order to check some differences in the simulated flow regimes among them.

Though some important modifications and corrections are required, my recommendation is accept after minor revision, considering that the revised paper has been improved, especially for the English style. Nevertheless, I strongly suggest the authors to carefully address all my comments.

Cited works:

[1]: https://doi.org/10.1016/j.jhydrol.2018.11.001

[2]: https://doi.org/10.1016/j.jhydrol.2019.06.031

[3]: https://doi.org/10.1016/j.jhydrol.2018.09.028

Reviewer 2 Report

This a review of a resubmitted manuscript entitled “Comparative Analysis of Building Representations in TELEMAC-2D for Flood Inundation in Idealized Urban Districts”. In my previous review reports I considered the manuscript unsuitable for publication because of (1) insufficient literature review; (2) missing the analysis of the model’s numerical accuracy. (3) lack of the discussion on the differences between the presented solution and observations.

In the present version of the manuscript, authors successfully addressed all these issues and I have only four minor remarks:

Lines 199-202 cm-resolution scales, are rather not general (large scale model with 1cm grid?). Please explain it in the text.

Line 261 – consider replacing “values” with “water levels”.

Line 279 – which simulation? Other simulations? Please clarify.

Section 3: consider switching the order: in my opinion the section 3 should start from the analysis of mesh resolution (now subsec. 3.3).

As I stated before, the study is valuable as presents the capabilities of the TELEMAC-2D model for the Toce River case study, modeled previously using other models.

Author Response

This manuscript is a resubmission of an earlier submission. The following is a list of the peer review reports and author responses from that submission.

Round 1

Reviewer 1 Report

The authors need to run this manuscript through a professional editor to correct the poor English grammar (evident even in the paper title) in this manuscript. I am unwilling to provide an exhaustive list of these issues.

Whilst the analysis of different building treatments is an important concern for urban flood modelling, the discussion of the results is not insightful enough for a journal paper.

Specific comments:

L101 What is TELAMC-2D?

EQ2&3 Replace "div" with proper symbol

L124 Do the authors have permission to use these pictures?

L211 "Obviously more matching" is too sloppy language for a paper.

Table1&2 These are hard to read. Can they be replaced with figures instead?

Author Response

Response to Reviewer 1 Comments

The authors need to run this manuscript through a professional editor to correct the poor English grammar (evident even in the paper title) in this manuscript. I am unwilling to provide an exhaustive list of these issues.

Whilst the analysis of different building treatments is an important concern for urban flood modelling, the discussion of the results is not insightful enough for a journal paper.

Specific comments:

L101 What is TELAMC-2D?

Response: Thanks for your comments. TELEMAC-2D is a powerful two dimensional hydrodynamic model, and it solves Saint-Venant equations based FEM or FVM. The more details about TELEMAC-2D could refer to the book “Hydrodynamics of free surface flows modelling with the finite element method” written by Hervouet, J.M or the website “http://www.opentelemac.org/”. The manuscript do some studies based on the TELEMAC-2D model and the introduction of this model is in section 2.1

EQ2&3 Replace "div" with proper symbol

Response: Thanks for your comments. The equations 2 and 3 are corrected with proper symbol.

L124 Do the authors have permission to use these pictures?

Response: Thanks for your comments. The copyright of these pictures is very important. The Toce River Valley case is a very famous case and these pictures have implemented in many literatures. In addition, we have made reasonable reference obviously in paper. I think there is not conflict.

L211 "Obviously more matching" is too sloppy language for a paper.

Response: Thanks for your comments. We are very sorry for the sloppy language. We have tried our best to improve the English language. Some poor English grammar and English expression have corrected.

Table1&2 These are hard to read. Can they be replaced with figures instead?

Response: Thanks for your comments. Table 1&2 describe the RMSE and PPMCC values to quantify the numerical model performance. We have tried some different types of figures to better understand the results, but it is still difficult to read and the figure would be not clear due to a lot of data. The two tables have presented the complete results accurately. And we could add more descriptions to better analyze and understand results

Reviewer 2 Report

The authors present a paper in which an existing hydrodynamic model (Telemac2D) is applied within an urban context to understand the efficacy of model use in more complex environments. The authors used the classic Tolce physical model as the single evaluation of the implementation 3 approaches to representing configurations of buildings- Building Hole, Building Block, and Building Resistance. Simulations using these configurations were compared to measurements at distinct locations. The authors note little difference between Telemachus 2D and the observations, thus stating that Telemac2D is useful in the context of urban flooding.

The research presented here is very simple. No new novel concepts were presented, except that perhaps the 3 approaches for representing buildings were not previously attempted within Telemac2D. That, in itself, could be useful. However, there is very little discussion about the implementation, trade offs in approaches relative to implementation (e.g., pre-processing, computational tradeoffs), and the statistical comparisons for each not thoroughly discussed. For example, in the context of urban flood prediction and damage assessment, what is the significance of building representation within Telemac2D? The results are not articulated in such a way that makes the research compelling.

I believe the concept has merit, but requires more attention to identification and justification of the research questions.

Author Response

Response to Reviewer 2 Comments

The authors present a paper in which an existing hydrodynamic model (Telemac2D) is applied within an urban context to understand the efficacy of model use in more complex environments. The authors used the classic Toce physical model as the single evaluation of the implementation 3 approaches to representing configurations of buildings- Building Hole, Building Block, and Building Resistance. Simulations using these configurations were compared to measurements at distinct locations. The authors note little difference between Telemachus 2D and the observations, thus stating that Telemac2D is useful in the context of urban flooding.

The research presented here is very simple. No new novel concepts were presented, except that perhaps the 3 approaches for representing buildings were not previously attempted within Telemac2D. That, in itself, could be useful. However, there is very little discussion about the implementation, tradeoffs in approaches relative to implementation (e.g., pre-processing, computational tradeoffs), and the statistical comparisons for each not thoroughly discussed. For example, in the context of urban flood prediction and damage assessment, what is the significance of building representation within Telemac2D? The results are not articulated in such a way that makes the research compelling.

I believe the concept has merit, but requires more attention to identification and justification of the research questions.

Response:

Thanks for your comments. We have studied your comments carefully.

The research target is to model the urban flood based on TELEMAC-2D and compare the three approaches for representing buildings, which are analyzed and discussed in the paper in detail. In addition, we find and discuss some interesting phenomenon that might be helpful for urban flood risk management. The setup of numerical model is described in detail and sensitive analysis of Manning Coefficient is supplemented. And the computational tradeoff is not the key purpose in this paper, therefore we don’t make more model tests to study, but we do some literature reviews to set up model parameters instead. We add some discussions about Manning Coefficient, model performance, Froude number and so on in order to enrich the paper and explain the model results comprehensively, especially gauges P5 in aligned and P8 in staggered. Also, we underline that the results of this paper might be helpful in urban flood simulation and management.

We would like to express our great appreciation to your comments.

Reviewer 3 Report

This is a benchmark-kind study on the usage of TELEMAC-2D model for urban flood flows. Authors have focused on the techniques for representing buildings in a two-dimensional numerical model, including: Building-Hole, Building-Block and Building-Resistance approaches. Computations were performed for laboratory data published by Testa et al. 2007 [17].

As the benchmark, the study is interesting, as it is a first application of TELEMAC-2D to this data set, but I am unsure about its general scientific value. Testa et al. 2007 dataset was used in numerous numerical studies, including different ways of building representation. The manuscript provides only short references to some of these studies (line 122, but i.e. very similar Dottori, F. and Todini, E. (2013), Testing a simple 2D hydraulic model in an urban flood experiment. Hydrol. Process., 27: 1301-1320. doi:10.1002/hyp.9370 is missing) and it is unsure how it adds anything new to the subject. This my main remark, I would like authors convince readers, that their approach is original.  I expect that they will provide a detailed analysis of other numerical results for this data set, including ways to represent buildings, and discuss it in the respect of their outcomes.

I have also doubts on the documentation of numerical experiments. There is no information on numerical accuracy of the solution. How the time step was chosen? What were min/max Courant numbers? What was the numerical accuracy of the solution? These issues are important, as previous studies using this dataset indicated a numerical stability problems (maybe not present with TELEMCA-2D), because of large bottom slopes (>0.01).

The important source of numerical accuracy is the discretization of the computational domain. As the model is quite simple and computational times are not very high, I think the analysis of the model sensitivity to the mesh resolution would allow to justify, how much the solution is affected by this factor (an example is given by: Hu, J. (2018). A simple numerical scheme for the 2D shallow-water system. arXiv preprint arXiv:1801.07441.).

Please note, that authors solution reveals some significant discrepancies with laboratory data. First of all, the peak travel times in the numerical solution are noticeably smaller than observed ones. It can be seen for almost all hydrographs, were observations are “delayed” by several seconds. What is the reason? The second issues is the hydraulic jump, which in laboratory data was present upstream the buildings (measuring points P3-4) , but in authors solution downstream the sections of P5. These diferencies should be indicated and somehow explained, referring to other numerical studies, and maybe numerical accuracy.

I am not a native English speaker, but although language of the text is easy to understand, it could benefit from some minor language polishing.

Bellow, please find some minor  remarks:

Line 106, I suggest more specific definition for ve, as “effective diffusion for turbulent viscosity and dispersion”

Line 183: 10^1 would not be sufficient? What is a justification for a such high Manning value?

Author Response

Response to Reviewer 3 Comments

This is a benchmark-kind study on the usage of TELEMAC-2D model for urban flood flows. Authors have focused on the techniques for representing buildings in a two-dimensional numerical model, including: Building-Hole, Building-Block and Building-Resistance approaches. Computations were performed for laboratory data published by Testa et al. 2007 [17].

As the benchmark, the study is interesting, as it is a first application of TELEMAC-2D to this data set, but I am unsure about its general scientific value. Testa et al. 2007 dataset was used in numerous numerical studies, including different ways of building representation. The manuscript provides only short references to some of these studies (line 122, but i.e. very similar Dottori, F. and Todini, E. (2013), Testing a simple 2D hydraulic model in an urban flood experiment. Hydrol. Process., 27: 1301-1320. doi:10.1002/hyp.9370 is missing) and it is unsure how it adds anything new to the subject. This my main remark, I would like authors convince readers, that their approach is original.  I expect that they will provide a detailed analysis of other numerical results for this data set, including ways to represent buildings, and discuss it in the respect of their outcomes.

Response: Thanks for your comments. TELEMAC-2D is a powerful 2-D hydrodynamic model and has been applied in dam break, storm surge, and harbor structure and so on. However, few researches have applied this model to simulate urban flood. The key purpose of this paper is to validate the usability of TELEMAC-2D in urban flood scenario. The Toce case is a widely used in many studies, and this paper use the case to provide a benchmark of TELEMAC-2D model in urban flood scenario. The results of this paper would be helpful to explore more applications about TELEMAC-2D model. In addition, we add some discussions about Manning Coefficient, model performance, Froude number and so on in order to enrich the paper and explain the model results comprehensively, especially gauges P5 in aligned and P8 in staggered. Also, we underline that the results of this paper might be helpful in urban flood simulation and management.

I have also doubts on the documentation of numerical experiments. There is no information on numerical accuracy of the solution. How the time step was chosen? What were min/max Courant numbers? What was the numerical accuracy of the solution? These issues are important, as previous studies using this dataset indicated a numerical stability problems (maybe not present with TELEMCA-2D), because of large bottom slopes (>0.01).

Response: Thanks for your comments. The setup parameters of numerical model are very important. The choices of time step and Courant number are supplemented. The max Courant number value of the six numerical model in this paper is 2.2 and the average CFL value is about 0.5. The choice of time step is explained in detail. And more descriptions about numerical experiments are in section 3.1.

The important source of numerical accuracy is the discretization of the computational domain. As the model is quite simple and computational times are not very high, I think the analysis of the model sensitivity to the mesh resolution would allow to justify, how much the solution is affected by this factor (an example is given by: Hu, J. (2018). A simple numerical scheme for the 2D shallow-water system. arXiv preprint arXiv:1801.07441.).

Response: Thanks for your comments. The 2 cm mesh resolution is accuracy enough to depict the Toce River Valley case according some literatures. And the computational domain is discretized by 187476 triangular elements with BlueKenue free software. And the computational tradeoff is not the key purpose, therefore we don’t make analysis about the influence of mesh resolution. But we do some more tests to analyze the influence of Manning Coefficient for buildings grid to enrich the paper.

Please note, that author’s solution reveals some significant discrepancies with laboratory data. First of all, the peak travel times in the numerical solution are noticeably smaller than observed ones. It can be seen for almost all hydrographs, were observations are “delayed” by several seconds. What is the reason? The second issues is the hydraulic jump, which in laboratory data was present upstream the buildings (measuring points P3-4), but in authors solution downstream the sections of P5. These differences should be indicated and somehow explained, referring to other numerical studies, and maybe numerical accuracy.

Response: Thanks for your comments. The two issues are very important and interesting to analyze. We have noted and studied. For the first question, the reason might be the grid resolution, especially related to the process of grids changed from dry to wet. For the second question, the reason might be concerned with Froude number, critical flow has some difficult to simulate. Related discussions are in section Discussion.

I am not a native English speaker, but although language of the text is easy to understand, it could benefit from some minor language polishing.

Response: Thanks for your comments. We have tried our best to improve the English language. Some poor English grammar and English expression have corrected.

Bellow, please find some minor remarks:

Line 106, I suggest more specific definition for ve, as “effective diffusion for turbulent viscosity and dispersion”

Response: Thanks for your comments. The definition of ve is corrected as your suggestion.

Line 183: 10^1 would not be sufficient? What is a justification for a such high Manning value?

Response: Thanks for your comments. The sensitivity analysis of Manning Coefficient is supplemented in section 3.2. And the conclusion is that it is sufficient when the Manning Coefficient is larger than 104, but it is set up 1010 for accuracy purpose in this paper.

Reviewer 4 Report

This paper deals with the capability of a well-known commercial software (TELEMAC-2D) for the simulation of flood propagation in urban districts.

In particular, the authors have considered experimental tests in which the urban area was idealized through a serie of blocks arranged according to a staggered or aligned configuration. 

The paper focuses on three different techniques to represent the effects of the urban area within the 2-D models.

The topic faced by the authors is of great importance and I think is of interest for the "Water" readers.

However, this research is not particular original since a lot of scholars have performed this kind of analysis in order to validate finite volume schemes and Riemann solvers for the numerical solution of the Shallow Water Equations. The only element of novelty is the analysis of the performances related to the use of TELEMAC-2D but, as the authors observed, it now uses a finite volume method with well-balanced properties. Therefore, this paper does not add significant information in respect to other simular works in the literature.

For this reason, though the paper has some interesting points, I think that should be significantly improved before to consider it again for possible publication.

In order to help the authors to provide a more interesting and novel analysis, I strongly suggest to address the following points:

1. Analysis of results: besides the analysis of the water depths levels, an original issue that the authors can discuss is the effect of the three techniques on the flow velocity. Though experimental data are not avaiable, I think that I deeper discussion on this aspect can give some element of novelty. Similarly, the authors can discuss the effects on the flow regime, considering for example the froude number computed in an instant of time just after the peak. 

2. Motivation of the work: the authors failed to underline the importance of the analysis of TELEMAC-2D in simulations like these. In order to improve this part I suggest to add something, in the introduction or in conclusion, about the role of commercial and widely tested numerical models within the flood risk management. In particular, it would be wise to connect their research with some novel issues discussed in the literature such as the use of this model for flood risk communication in urban areas (see for example [1,2]). In this way, the authors can extend the field of application of their research to a more wide and novel topic.

3. The title is too ambitious and does not reflect the content of the work. The authors have not simulated real-world urban floods but simply reproduced experimental tests. Moreover, the core of the work is the use of TELEMAC-2D. So, a more suitable title could be: Comparative analysis of buildings representation in TELEMAC-2D for flood inundation in idealized urban districts (or something like this).

4. I think it should be necessary to mention all the studies that provide simulations for these test cases. For example, but the list is not complete, you can consider [3,4,5,6]. It essential to underline what is novelty introduced in respect to other papers that performed a similar analysis. 

5. Line 182: the manning coefficient for buildings grid was set equal to 10^10. This value is quite strange. A deeper discussion on this should be added and a sensitivity analysis probably should be performed.

6. What about the Courant number? Is it the same for all the simulations? some comment for the total number of cells.

7. I'm not an English native speaker but I think that a revision of the English style is necessary.

cited works

[1]:DOI: 10.1016/j.envsoft.2018.11.005

[2] DOI 10.1007/s11027-015-9651-2

[3] DOI: 10.1002/hyp.9370

[4]DOI: 10.1016/j.advwatres.2011.11.002

[5]DOI: 10.1016/j.jher.2012.01.001

[6]DOI: 10.1016/j.envsoft.2015.01.009

Author Response

Response to Reviewer 4 Comments

This paper deals with the capability of a well-known commercial software (TELEMAC-2D) for the simulation of flood propagation in urban districts.

In particular, the authors have considered experimental tests in which the urban area was idealized through a series of blocks arranged according to a staggered or aligned configuration.

The paper focuses on three different techniques to represent the effects of the urban area within the 2-D models.

The topic faced by the authors is of great importance and I think is of interest for the "Water" readers.

However, this research is not particular original since a lot of scholars have performed this kind of analysis in order to validate finite volume schemes and Riemann solvers for the numerical solution of the Shallow Water Equations. The only element of novelty is the analysis of the performances related to the use of TELEMAC-2D but, as the authors observed, it now uses a finite volume method with well-balanced properties. Therefore, this paper does not add significant information in respect to other similar works in the literature.

For this reason, though the paper has some interesting points, I think that should be significantly improved before to consider it again for possible publication.

Response: Thanks for your response. Because it is not the key purpose to validate FVM and Riemann solvers in this paper, we don’t add many literatures in this paper. TELEMAC-2D solves Shallow Water Equations based on FEM or FVM, where FEM is the one of the most important character and FVM is added later in 6th version. This paper use FEM to simulate the Toce River Valley case and more details about model setup are described in section 3.1. In addition, it is interesting to analyze the difference performance between FVM and FEM, which is the next study in the future.

In order to help the authors to provide a more interesting and novel analysis, I strongly suggest to address the following points:

1. Analysis of results: besides the analysis of the water depths levels, an original issue that the authors can discuss is the effect of the three techniques on the flow velocity. Though experimental data are not available, I think that I deeper discussion on this aspect can give some element of novelty. Similarly, the authors can discuss the effects on the flow regime, considering for example the Froude number computed in an instant of time just after the peak.

Response: Thanks for your response. Because computed velocities could not be assessed due to the absence of observed data, the Froude number is analyzed in this paper, especially the instant of time just after the depth peak in section Discussion. And we analyze the P5 in aligned and P8 in staggered based on Froude number in detail.

2. Motivation of the work: the authors failed to underline the importance of the analysis of TELEMAC-2D in simulations like these. In order to improve this part I suggest to add something, in the introduction or in conclusion, about the role of commercial and widely tested numerical models within the flood risk management. In particular, it would be wise to connect their research with some novel issues discussed in the literature such as the use of this model for flood risk communication in urban areas (see for example [1,2]). In this way, the authors can extend the field of application of their research to a more wide and novel topic.

Response: Thanks for your comments. Few researches have applied TELEMAC-2D model to simulate urban flood and it is very important to extend the field of application. We add some more discussions about flood management in section Discussion in order to enrich the paper. And it is also the next step to study urban flood risk and damage assessment.

3. The title is too ambitious and does not reflect the content of the work. The authors have not simulated real-world urban floods but simply reproduced experimental tests. Moreover, the core of the work is the use of TELEMAC-2D. So, a more suitable title could be: Comparative analysis of buildings representation in TELEMAC-2D for flood inundation in idealized urban districts (or something like this).

Response: Thanks for your comments. A title is of great important in a paper. The proper title is adopted to better reflect our research after our careful consideration and the new title is “Comparative Analysis of Buildings Representation in TELEMAC-2D For Flood Inundation in Idealized Urban Districts” as your suggestion.

4. I think it should be necessary to mention all the studies that provide simulations for these test cases. For example, but the list is not complete, you can consider [3,4,5,6]. It essential to underline what is novelty introduced in respect to other papers that performed a similar analysis.

Response: Thanks for your comments. We have tried our best to mention all important studies in this case and the literatures you suggested has been noted and referred in this paper. The introduced novelty in this paper is underlined in section Result and Discussion, which deepens our understanding about the numerical model performance in urban district. Also, we underline that the results of this paper might be helpful in urban flood simulation and management.

5. Line 182: the manning coefficient for buildings grid was set equal to 10^10. This value is quite strange. A deeper discussion on this should be added and a sensitivity analysis probably should be performed.

Response: Thanks for your comments. The sensitivity analysis of Manning Coefficient is supplemented in section 3.2. And the conclusion is that it is sufficient when the Manning Coefficient is larger than 104, but it is set up 1010 for accuracy purpose.

6. What about the Courant number? Is it the same for all the simulations? Some comment for the total number of cells.

Response: Thanks for your comments. The setup parameters of numerical model are very important. The choices of time step and Courant number are supplemented. The max Courant number value of the six numerical model in this paper is 2.2 and the average CFL value is about 0.5. The choice of time step is explained in detail. And more descriptions about numerical experiments are in section 3.1.

7. I'm not an English native speaker but I think that a revision of the English style is necessary.

Response: Thanks for your comments. We are very sorry for the poor language. We have tried our best to improve the English language. Some poor English grammar and English expression have corrected.

Round 2

Reviewer 3 Report

My main remark in the previous review was a lack of the originality in the presented approach. The study is based on a rather known case study, which was analyzed numerically several times. I asked authors to explain what is their input to the problem of modeling urban floods. In my opinion, in their revision they did not succeed to show any important input to the present knowledge. First of all, the state-of-the-art, even for Toce case study is still missing. In the revision, authors included references to other studies (only in the discussion/result section) without any deeper analysis of their findings.

The only novel element of the study is the use of TELEMAC-2D. Such benchmark can be scientifically valuable, only with detailed analysis of a model convergence and sensitivity (i.e. for mesh resolution). However, the manuscript provides information only on fit measures. This is not enough for a benchmark. Especially, the fit is far from being perfect - faster peak in the numerical model (btw authors explain it i.a. with insufficient mesh resolution, but without mesh sensitivity analysis it is just a guess) or “oscillations” in the computed hydrographs (Can we be sure it is not a numerical issue?) .

The analysis of the value of the Manning coefficient to represent the buildings cannot be considered as an original element. It is just a technical issue. Only interesting thing is, why authors insist on value of 10^10, if 10^4 is sufficient.

Reviewer 4 Report

This is my second review of this paper.

I've appreciated the efforts made by the authors to take into account my and other suggestions, even though some remarks have not been fully addressed by the authors.

In general, the manuscript has benefited by the review process but I don't think it is yet ready for publication.

My main concerns are listed below.

the work still has a lack of focus. The authors should  underline better the added value of their research according to previous studies in the literature. In particular, if the goal is the analysis of TELEMAC-2D in urban areas, they should explain why they consider this issue so important.

I still have some doubts on the analysis performed by the authors related to the Manning coefficient values. In particular, what does figure 6 mean? What is this "Unit width discharge in buildings grid". Why it is so important for the authors? I think that is obvious to check the influence of the Manning coefficient on the observed water depth values. So please, add this kind of analysis in your study. 

Linked to point number 1, I think that the authors should deepen the analysis of the results. For this reason, though flow velocity data are not available, it would be interesting show the simulated velocity profile simulated by the three methods considered in this work for building representation. It is interesting to evaluate the effect of these methods on flow velocity. Similarly, the Froude number analysis has a real meaning in this paper if the authors provide a comparison between these three approaches. Only in this way, in my view, they can discuss something interesting in their work.

Finally, the manuscript requires a thorough re-read to eliminate grammatical errors and, also, resolve unclear statements. I think that is necessary that an English native-speaker proof read the document.

Therefore, although the revised paper is improved, I recommend that the manuscript is returned to the Authors for major revisions.